# STRUCTURED WORLD MODELS FROM LOW-LEVEL OBSERVATIONS

## ABSTRACT

We present Structured World Modeling From Low-Level Observations ("SWMPO"), a framework for the unsupervised learning of neural Finite State Machines (FSM) that capture environment structure. Traditional unsupervised world modeling methods for policy optimization rely on unstructured representations, such as neural networks, which do not explicitly represent high-level patterns within the system (e.g., *walking* vs *swimming*). In contrast, SWMPO explicitly models the environment as an FSM, where each state represents a region of the environment's state space with distinct dynamics, exposing the structure of the environment to downstream tasks such as policy optimization. Prior works that synthesize FSMs for this purpose have been limited to discrete spaces, not continuous, high-dimensional spaces. Our FSM synthesis algorithm operates in an unsupervised manner, leveraging low-level features from unprocessed, non-visual data, making it adaptable across various domains. We demonstrate the advantages of SWMPO by benchmarking its environment modeling capabilities in different simulated environments.

## 1 INTRODUCTION

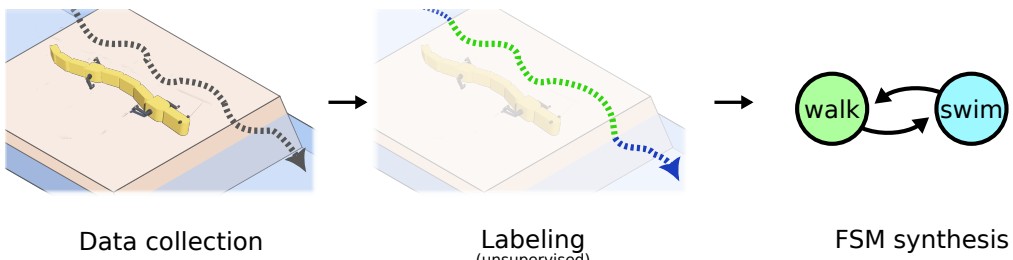

Figure 1: Overview of the proposed method. First, an existing (possibly expensive to run) controller (e.g., a planner) is used to gather data. Then, data is labeled according to the modes of the system in an unsupervised fashion. With this, a model of the environment could be used the form of a state machine is synthesized. In this illustration, the walking mode is green and the swimming mode is blue.

This paper examines learned approximations of environment dynamics, known as *world models* Ha & Schmidhuber (2018) in the special case where these models must explicitly encode the high-level structure of the environment dynamics. We are motivated by the observation that the high-level structure of a dynamical system can be used to efficiently solve control problems. Consider, for example, an amphibious robot that must navigate both water and land (see fig. 1). An expert roboticist might approach this problem by breaking down the task into three sub-problems: (1) controlling the robot on water; (2) controlling it on land and; (3) managing the transition between these two modes. With this division, the expert can exploit the fact that the robot moves faster on land than in water, using this knowledge to optimize route planning. Inspired by this strategy, our goal is to develop a method that automatically constructs a representation of the environment and a corresponding fine state machine.

We are interested in extracting structure directly from continuous, low-level, non-visual observations (e.g., LiDAR measurements or joint positions). To this end, we propose Structured World Modeling From Low-Level Observations –SWMPO– a framework where an environment's high-level structure is inferred directly from low-level continuous observations in a fully unsupervised manner (i.e., with no training labels), resulting in an FSM which can then be utilized in downstream tasks such as policy optimization.

The synthesized FSM consists of modes and transitions. Each mode is a neural network that approximates the environment dynamics within a subset of the state space (e.g., the *walking* mode, see fig. 1). Predicates determine when to switch between modes based on observations of the environment. We evaluate SWMPO across a variety of benchmarks and environments with continuous dynamics, including two-and three-dimensional simulations.

**Contributions**   Our contributions are as follows:

1. A novel unsupervised learning algorithm that segments time series data into a discrete set of modes.
2. A state-machine synthesis algorithm that constructs a Finite State Machine (FSM) model directly from continuous low-level observations, enabling interpretable representations of latent dynamics.
3. Empirical testing demonstrating the performance of the state machine across fours test environments.

## 2 RELATED WORK

**Automata Synthesis and Symbolic Structure Extraction**   Hasanbeig et al. (2021) demonstrated that FSMs could be synthesized to model environments, improving performance in RL tasks. However, this method is limited by its reliance on fully-symbolic representations obtained from pre-trained vision models in grid-world settings. In contrast, our approach extracts structure from continuous, low-level, non-visual observations. To the best of our knowledge, our work is the first to leverage neural world models in the synthesis of FSMs for continuous low-level high-dimensional non-visual observation spaces.

**Hidden Markov Models**   Hidden Markov Models (HMMs) are a standard approach to capturing temporal dependencies and mode-switching behavior in sequential data (Li & Biswas, 2002; Bouguila et al., 2022). In robotics, HMMs have been leveraged to segment trajectories into discrete modes (Goh et al., 2012) and used during policy learning for multimodal or hierarchical tasks (Marturi et al., 2019). Recent advances have extended HMMs using deep neural network architectures (neural HMMs) to handle high-dimensional, continuous observation spaces and to learn more complex transition dynamics (Tran et al., 2016). For instance, neural HMMs have been used in unsupervised settings to model complex sensory streams for trajectory clustering Vakanski et al. (2012) and to predict latent modes during task execution Wu et al. (2019). However, HMMs suffer from the fundamental limitation that the transition between modes is determined by a probability distribution that is only conditioned on the latent state, this means that the observed evolution of the system itself is only indirectly used to update the active mode.

**Leveraging Structure in Reinforcement Learning**   A body of research focuses on leveraging structure to solve control problems with RL (Mohan et al., 2024). Hierarchical RL (Xu & Fekri, 2021; Botvinick, 2012; Li et al., 2006) and modular RL (Simpkins & Isbell, 2019; Andreas et al., 2017; Devin et al., 2017) encode structure directly into the policy architecture, here we instead consider the synthesis of a structured model. Model-based RL approaches leverage neural world models to optimize policies more efficiently (Moerland et al., 2023; Ha & Schmidhuber, 2018). However, neural models lack distinct boundaries between the representation of different modes in the environment. Reward machines (Toro Icarte et al., 2019; Icarte et al., 2018) leverage structured models of the reward function to guide the policy optimization process.

**Structure Induction and Hybrid Systems**   In the broader field of hybrid systems, modeling environments as a collection of modes with distinct dynamics is standard practice (Alur et al., 1995;

Paoletti et al., 2007; Ferrari-Trecate et al., 2003; Devin et al., 2017; Camacho et al., 2010). Recently, Soto et al. (2021) show that automata with affine dynamics can be synthesized from time-series data. In this work, we leverage non-linear neural models to both extract structure and represent dynamics within the automata, making our approach more general.

Finally, other approaches to discovering structure which are not directly applicable to time-series data include the use of graph neural networks (Cranmer et al., 2020) and sparse networks (Gupta et al., 2024). Similarly, methods that leverage recurrent neural networks (RNNs) and FSMs to model linguistic structures (Kolen, 1993; Koul et al., 2018; Jacobsson, 2005) share a conceptual foundation with our work, but these methods focus on formal languages and are not directly applicable to the class of problems we study in this work.

## 3 BACKGROUND

We operate in the standard discrete-time RL framework, where an agent interacts with an environment (Sutton & Barto, 2018). A summary of the notation used in this paper can be found in table 1.

### 3.1 REINFORCEMENT LEARNING

**Definition 1** (Partially Observable Markov Decision Process). A discrete-time Partially Observable Markov Decision Process (POMDP) is a tuple $\mathcal{M} \triangleq \langle S, A, T, S_0, \Omega, O \rangle$, where $S \in \mathbb{R}^k$ is the set of states, $\Omega$ is the set of observations that the agent can make, $A$ is a set of actions, $T : S \times A \to S$ is a transition function, $S_0$ is a distribution of initial states, $\Omega$ is a set of observations, each observation $o \in \Omega$ made under some state $s \in S$ and action $a \in A$ with probability $O(o \mid s, a)$ given by the set of conditional observation probabilities. We associate a POMDP with a reward function $R : S \times A \times S \to \mathbb{R}$.

**Definition 2** (Trajectory). A trajectory is a time-indexed sequence of transition tuples $(o_t, a_t, o_{t+1})$. We call $o_t$ and $o_{t+1}$ the source and next observations, respectively.

### 3.2 FINITE STATE MACHINES

**Definition 3** (Finite State Machine World Model). We define a finite state machine world model (FSMWM) to be a tuple $\mathcal{F} \triangleq \langle f, O, A, \delta, f_0 \rangle$, where $O$ and $A$ are respectively the observation and action spaces of a POMDP; $f = \{f_i : O \times A \to O\}_i$ is a set of models of the environment; and $\delta = \{\delta_{i,j} : O \times A \to \{0, 1\}\}_{i,j}$ is a set of mode-transition predicates.

For a given active mode indexed by $i$ and an observation $o_t \in S$, the predicted next observation is $o_{t+1} = f_i(o_t, a_t)$ is the predicted next observation. The next active mode index is

$$\delta(o_t, u_t, i) = \begin{cases} \operatorname{argmax}_j \delta_{i,j}(o_t, u_t) & \text{if } \delta_{i,j}(o_t, u_t) > 0 \\ i & \text{otherwise.} \end{cases}$$

We define $\operatorname{argmax}$ to choose the first matching index in case of a tie. This definition mirrors previous use of a FSM as policies in the MDP setting (Inala et al., 2020), but here we are using them as world models.

## 4 PROBLEM STATEMENT

Consider the example of an amphibious robot that must navigate both water and land (see fig. 1), corresponding to the two modes of the system. Our goal is two-fold: (1) to synthesize an FSMWM from low-level observations in an unsupervised manner (i.e., without mode labels) that captures the high-level structure of the environment by moving between modes that correspond to the (unobserved) mode of the system, and (2) to leverage the FSMWM in the RL training loop.

Our fundamental assumption is that the latent categorical variable $M_t$, which corresponds to the modes, can be characterized by a function $m : (s_{t-1}, a_{t-1}, s_t) \mapsto m_t$. Further assumptions made by our method on the system and $M_t$ are motivated and stated in section 5.1.

## 5  METHOD

This section outlines the two main components of the SWMPO algorithm: (1) State Machine Synthesis, where we use data from episodes to synthesize an FSM that models the environment's structure and low-level agent behavior, and (2) State-Machine-Guided Policy Optimization, where the synthesized FSM is employed to optimize the policy. Our framework is outlined in algorithm 1.

---

**Algorithm 1** SWMPO

---

**Require:** POMDP $\mathcal{M} = (S, A, T, S_0, \Omega, O)$, initial policy $\pi_0$, reward function $R$, mode number $m$,
       partition pruning tolerance error $\epsilon$, learning rate $\gamma$, intrinsic reward factor $\eta$, RL algorithm
 1: Collect trajectory dataset $D$ with $\pi_0$
 2: $\mathcal{F} = \mathsf{synthesizeFSMWM}(D, \pi_0, m, \epsilon, \gamma)$
 3: **return** $\mathcal{F}$

---

As illustrated in fig. 1, the inputs to our proposed framework, SWMPO, are a POMDP $(S, A, T, S_0, \Omega, O)$ with associated reward function $R$, the number of modes $m$, and an expert policy $\pi_0$ used for initial data collection. The outputs are (1) a state machine with $m$ nodes that approximates $T$, and (2) a policy that approximately maximizes the reward in the POMDP.

Our method synthesizes an FSM to model the structure of the environment, where each state in the FSM represents a distinct mode of the data (e.g., *swimming* or *walking*). A key challenge is discovering these modes in an unsupervised manner, as only the number of modes is assumed to be known. Ensuring clear separation between modes at different stages of the algorithm is also critical to avoid cascading errors from misclassified data, which can progressively degrade the model's performance. To address these challenges, we synthesize our state machine as follows:

1. **Labeling**: Divide the transitions in the dataset into different mode subsets.
2. **Pruning**: Simplify the mode-transition dynamics of the partition by removing spurious transitions between modes.
3. **Transition Predicate Synthesis**: Learn when to transition between modes.

### 5.1  LABELING

Labeling addresses the problem of decomposing environment dynamics by assigning each transition in a dataset $D$ of trajectories to one of $m$ disjoint subsets, with each subset corresponding to a mode of the POMDP. We first state the assumptions of the labeling algorithm, and then describe the algorithm.

Let $\langle S, A, T, S_0, \Omega, O \rangle$ be a POMDP. Let $S_t$ and $A_t$ be the random variables of the state and action at time $t$ respectively, under some fixed policy $\pi$. We focus on the case where observations conditioned on a state are deterministic, so $o_t = O(s_t)$. We are interested in modelling the mode variable $M_t$ (e.g., $m_t = walking$), taking values in some set $M$. Our method revolves around learning $M_t$ as an intermediate computation of a learned first-order model of the form

$$f(m(o_{t-1}, a_{t-1}, o_t), o_t, a_t) \approx o_{t+1} - o_t$$

which we describe in this section. Ultimately we characterize modes as a categorical variable (i.e., robot is either *walking* or *swimming*), but we first approximate $M_t$ as taking values in $\mathbb{R}^n$. We now impose constraints on the POMDP and the mode variable that allows us to design an algorithm to predict $M_t$. In summary, the assumptions imply that partial observability is a consequence solely of the latent mode variable and that this variable can in principle be predicted from previous observations.

**Assumption 1: mode identifiability**    We start with the assumption that $M_t$ can be modelled as a function $m_t \approx m(o_{t-1}, a_{t-1}, o_t)$. Intuitively, this assumption means that it is possible to identify the current mode by observing how the world changed under the latest action. We thus think of $M_t$ as an abstraction over the observed change of the system under some action and state.

**Assumption 2: change can be predicted conditioned on mode** The next assumption is that the POMDP becomes a deterministic MDP conditioned on the mode. More precisely, we assume the existence of a function $T' : M \times A \times O \to O$ such that $T'(m_t, a_t, o_t) = O(T(s_t, a_t))$.

Thus far, our constraints allow the trivial solution $M_t = S_t$, which is not useful. There may be many other variables which satisfy our assumptions. Consequently, we add a constraint that allows us to uniquely identify $M_t$.

**Assumption 3: modes alone have minimal information** Let $\mathbf{M}$ be the set of random variables that satisfy the previous assumptions. Then the mode variable $M_t$ is the unique solution to $M_t = \arg\min_{M_t \in \mathbf{M}} I(M_t, O_{t+1})$.

Under the assumptions so far, we can so far conclude that if we find a variable $M_t$ that allows us to predict the change in the environment given an action and has minimal mutual information with $O_{t+1}$, then $M_t$ must be the mode variable. However, our approximation of $M_t$ takes values in a vector space, but we ultimately want to model it as a categorical variable. We therefore add our last assumption.

**Assumption 4: mode vectors form clusters** $M_t$ corresponds to a partition of the state space where in expectation the within-subset sum of squares to the centroid of the subset is minimal. That is, we assume a *strict partitioning*, *centroid model* clustering scheme, which means that if $k$-means is run on vectors of $M_t$, then in expectation the clusters will correspond to the different modes.

Our assumptions imply that if a variable $M_t$ is predictive of the change in observed state for any action and has minimal mutual information with $O_{t+1}$, then that variable corresponds to the mode variable. In other words, consider $m : O \times A \times O \to M$ and $f : M \times A \times O \to O$ under the joint optimization problem

$$\arg\min_{e,d} \mathbb{E}_{(s_{t-1}, a_{t-1}, s_t, a_t, s_{t+1}) \sim \mathcal{M}_\pi} \left[ \| f(m(o_{t-1}, a_{t-1}, o_t), a_t) - (T(s_t, a_t) - s_t) \| \right] \tag{1}$$
$$- I(m(O_{t-1}, A_t, O_t), O_{t+1}),$$

where $\| \cdot \|$ is the Euclidean norm, $o_{t-1} = O(s_{t-1})$ and $o_t = O(s_t)$. From the assumptions stated above, it follows that the solution to eq. (1) implies that $m(\cdot)$ corresponds to the mode variable, which can be clustered with $k$-means to obtain mode labels for a set of data. In practice, we parametrize $m(\cdot)$ and $f(\cdot)$ with neural networks and approximate the solution with gradient-based search. To compute the mutual-information $I(\cdot, \cdot)$, we assume independence of features and fit Gaussian distributions to compute a Monte-Carlo approximation.

**Fitting local models to the data** At this point it is possible to fit a local model for each cluster of transitions in the dataset, as illustrated in Fig. 2, where each local model has higher performance for a particular mode of the environment. This entire process results in algorithm 2.

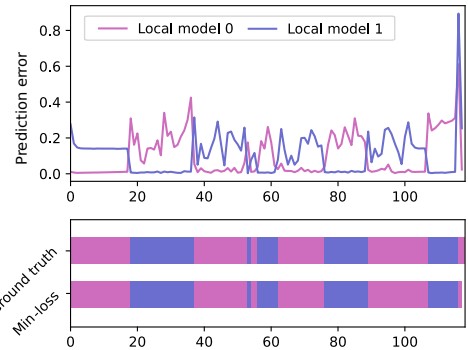

Figure 2: Performance of the specialized models for the *walking* and *swimming* modes of `PointMass`, an idealized version of an amphibious robot (see section 6). Each local model is specialized for a specific mode, leading to a combined low prediction error across the entire episode. The x-axis indicates time.

---

**Algorithm 2** optimizePartition

---

**Require:** list of trajectories $D$, latent space $M$, number of modes $m$
1: Use gradient-descent to find $e : S \times A \to M$ and $d : M \times A \to \hat{S}$ that approximately solve eq. (1)
2: Embed the transitions into mode vectors, $m(D)$
3: Cluster the embeddings into $m$ disjoint subsets using $k$-means
4: $D_i = \{\tau \in D \mid \text{cluster}(\tau) = i\}$
5: $\bar{D} = \{D_1, \dots, D_m\}$
6: Fit a local model $f_i$ to approximate the dynamics of $D_i$.
7: **return** $\bar{D}, f$

---

The algorithm takes a dataset of trajectories $D$, and partitions it by solving eq. (1) and clustering the resulting mode vectors. The partition, say $\bar{D} = \{D_1, \dots, D_m\}$, induces the sequence of modes that the state machine should visit for a given trajectory in the dataset. That is, the state machine should be in state $i$ when processing transition $\tau$ if and only if $\tau \in D_i$.

## 5.2 PRUNING

The aforementioned partitioning process can create overly complex transitions. While the FSM globally approximates the environment dynamics, some state regions may have multiple models with similar accuracy, resulting in spurious transitions between states. In such cases, transitions between these models can be pruned with minimal impact on performance.

To address this, we apply a pruning mechanism to eliminate these unwanted transitions. This helps balance the complexity-accuracy trade-off in the state machine search space: while more complex transition patterns can improve accuracy, they also increase the risk of overfitting and reduce interpretability. We now describe the pruning approach, which optimizes for both accuracy and simplicity.

**Pruning Approach**   We begin by labeling each transition in the dataset with the index of the neural network from the ensemble that best predicts the system's evolution in that state. A mode transition occurs when this label changes between consecutive states. For example, in the sequence `113322`, we transition from mode 1 to 3, then from 3 to 2. Pruning the transition to mode 3 yields two possible sequences: `111122` (forward-prune) or `112222` (backward-prune). Our goal is to remove transitions that have minimal impact on prediction accuracy.

To prune a mode transition, the framework shifts the affected transitions from one subset to another, causing a different model, with equal or greater prediction error, to handle those transitions. If the increase in prediction error is within the user-defined tolerance factor $\epsilon$, the move is considered $\epsilon$-valid relative to the original partition. A mode transition is $\epsilon$-prunable if all the associated moves are $\epsilon$-valid. There may be multiple $\epsilon$-prunable mode transitions for a given trajectory and partition. Our approach is to greedily prune the first prunable mode transitions with the strategy that results in the smallest prediction error increase (see algorithm 3).

---

**Algorithm 3** greedyPrune

---

**Require:** Partition $\bar{D}$ of trajectory dataset $D$, error tolerance factor $\epsilon$,
1: $\bar{D}_0 = \text{copy}(\bar{D})$
2: **for** $t \in D$ **do**
3:    **while exists** $\epsilon$-prunnable (relative to $\bar{D}_0$) mode transition in $t$ **do**
4:       Prune the first $\epsilon$-prunable (relative to $\bar{D}_0$) mode transition in $\bar{D}$, updating $\bar{D}$
5: **return** $\bar{D}$

---

### 5.2.1 TRANSITION PREDICATE SYNTHESIS

We describe the mechanism by which the FSM learns when to transition from one mode to another.

Each subset of a partition corresponds to a state of the FSM being synthesized. For each pair of FSM states $(f_i, f_j)$, the core question is: given that the state machine was in state $f_i$, and the agent observed $s_t$, took action $u$, and then observed $s_{t+1}$, should the FSM transition to state $f_j$? We identify the subset of $D_i$ containing transitions where the next state is a source state in $D_j$, referred to as the positive' set. The negative' set is the complement of the positive set with respect to $D_i$. The task then becomes a standard classification problem, where we find a predicate that outputs True for the positive set and False for the negative set. See algorithm 4. We use scikit-learn Pedregosa et al. (2011) to synthesize these predicates, parametrizing them with small Multi-Layer Perceptrons.

---

**Algorithm 4** synthesizeTransitions

---

**Require:** Partition $\bar{D} = \{D_1, \ldots, D_m\}$ and corresponding list of trajectories $D$
1: **for** $i, j \in \{1, \ldots, m\} \times \{1, \ldots, m\}$ **do**
2: $\quad$ positive $= \{\tau_1 \in D_i \mid \exists \tau_2 \in D_j \text{ s.t. follows}(\tau_1, \tau_2)\}$
3: $\quad$ negative $= D_i \setminus \{\text{positive}\}$
4: $\quad$ $\delta_{i,j} = \text{synthesizePredicate}(\text{positive}, \text{negative})$
5: **return** $\delta$

---

**Algorithm 5** synthesizeFSMWM

---

**Require:** Dataset $D$ of environment transitions, initial policy $\pi_0$, mode number $m$, partition pruning tolerance error $\epsilon$, learning rate $\gamma$, RL algorithm
1: $(\bar{D}', f') \triangleq \text{optimizePartition}(D, m, \gamma)$
2: Sort $\bar{D}' = \{D_1, \ldots, D_m\}$ so that $D_1$ contains the most initial transitions.
3: $\bar{D} = \text{greedyPrune}(\bar{D}', \epsilon, D)$
4: $\delta = \text{synthesizeTransitions}(\bar{D})$
5: $\mathcal{F} = (f, S, A, \delta, f_1)$
6: **return** $F$

---

## 6 EXPERIMENTS

We evaluate SWMPO's ability to identify and approximate the modes of the environment.

### 6.1 TEST ENVIRONMENTS

We test SWMPO on four environments of varying complexity (see fig. 3):

1. `PointMass`. These tasks are a simplified version of the amphibious robot running example, and consist of applying a sequence of thrusts to a two-dimensional point mass to take it to a target position. Crucially, the environment is split into terrains with different characteristics: sand with no drag and water with high drag. Additionally, to simulate the need for different policies in different terrains, actions in the sand terrain are inverted. See fig. 3a. We use an MPC controller as the initial expert policy.

2. `LiDAR-Racing`. Adapted from Ivanov et al. (2021). Tasks in this environment consist of driving a two-dimensional vehicle with bicycle dynamics and LiDAR sensors through a track randomly assembled from pieces of five different types. See fig. 3b. We use a pre-trained controller provided by the authors as the expert controller.

3. `Salamander`. A locomotion task in which an amphibious salamander must navigate through water and land. This environment is implemented in the Webots 3D simulator (Michel, 2004), in which the *Salamandra Robotica II* (Crespi et al., 2013) robot is available. See fig. 3c. This environment is a scaled-up version of `PointMass`. For observations, we use the motor positions, the LiDAR readings and the GPS position. We use the controller provided by Webots for this robot as the expert policy. However, to satisfy Assumption 2 we randomly switch the controller's mode, so that the robot sometimes performs swimming actions on the land and viceversa. This is so that the change in the world can only be accurately predicted if the mode variable is extracted.

4. `BipedalWalkerHarcore` (`BipedalWH`). A locomotion task in which a bipedal robot has to locomote over uneven terrain with four different types of obstacles. This is a standard benchmark in the Gymnasium library Towers et al. (2024) and employs the Box2D rigid body simulator Catto (2024). We use a pre-trained controller from the RL Baselines3 Zoo library Raffin (2020) as the expert policy.

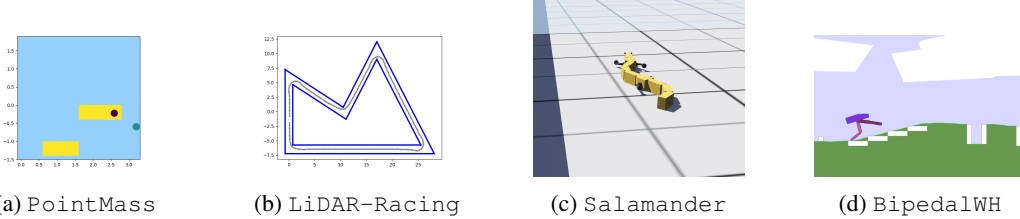

(a) `PointMass`        (b) `LiDAR-Racing`        (c) `Salamander`        (d) `BipedalWH`

Figure 3: Benchmark domains.

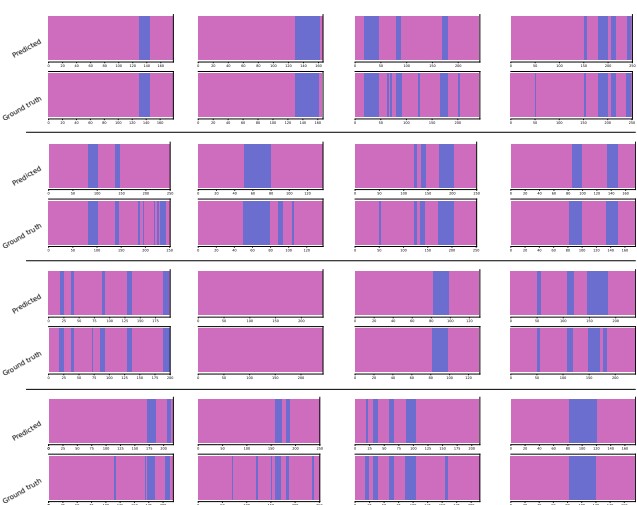

Figure 4: Labels of all the input trajectories from the `PointMass` environment. In each plot, the

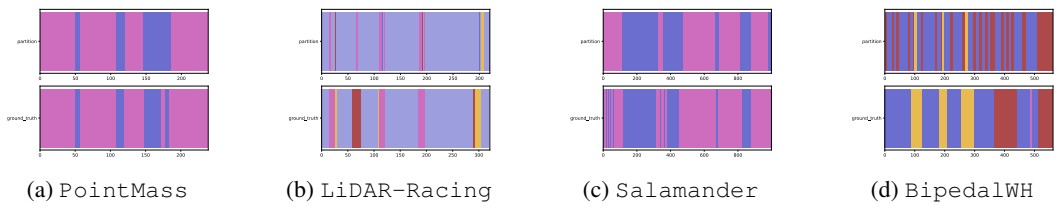

(a) `PointMass`        (b) `LiDAR-Racing`        (c) `Salamander`        (d) `BipedalWH`

Figure 5: Example unsupervised labeling outputs.

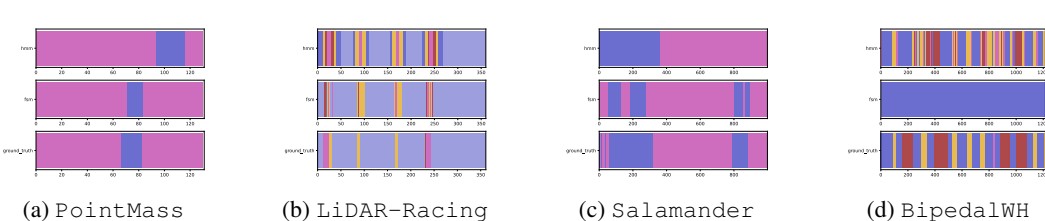

(a) `PointMass`        (b) `LiDAR-Racing`        (c) `Salamander`        (d) `BipedalWH`

Figure 6: Example mode tracking on unseen data.

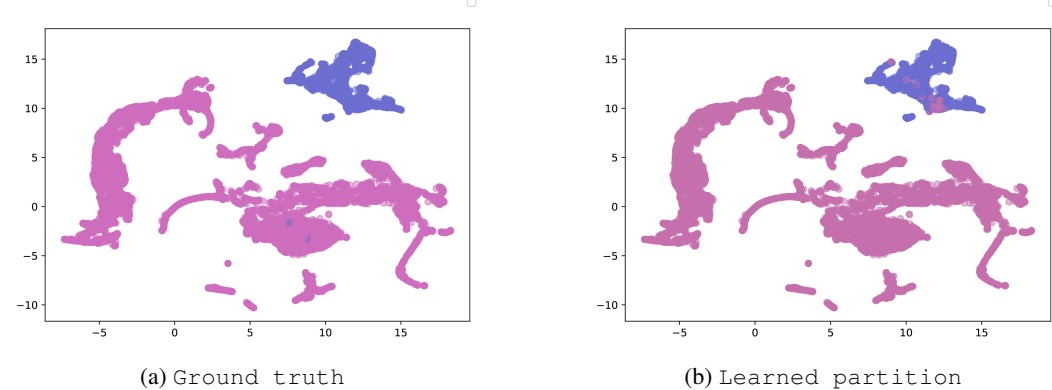

(a) Ground truth           (b) Learned partition

Figure 7: Mode vectors learned through SWMPO in the `PointMass` environment. Each point represents a transition encoded with the learned $m(\cdot)$ (after dimensionality reduction through UMAP). In the left plot, the ground truth labels are used to color the vectors. In the right, the learned partition is used to assign colors.

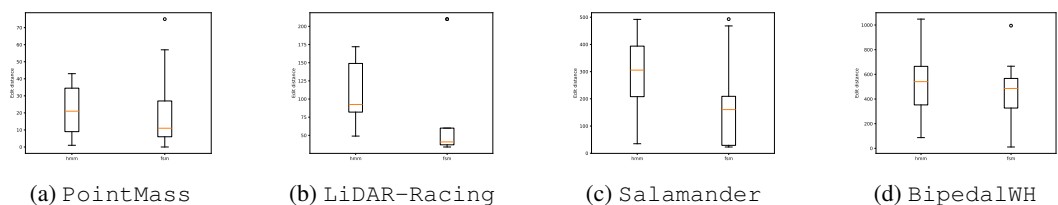

(a) `PointMass`     (b) `LiDAR-Racing`     (c) `Salamander`     (d) `BipedalWH`

Figure 8: We compare the performance of our method against HMMs across four different environments. The box plots illustrate the Levenshtein distance between FSM-predicted and ground truth labels for each environment, with SWMPO results shown on the right and HMM results on the left.

The algorithm to approximate the solution to eq. (1) is written with Pytorch (Ansel et al., 2024) using the Adam optimizer (Kingma & Ba, 2017). We use multi-layer perceptrons with ReLU activations for all the neural networks involved in the algorithm. Predicate synthesis is performed with Scikit-learn (Pedregosa et al., 2011).

## 6.2 LEARNED REPRESENTATION OF THE MODE VARIABLE

We present a plot of the data in the `PointMass` environment, where each transition is colored according to both the ground truth and the learned partitions, shown in fig. 7. The results indicate that the data forms clusters that align with the ground truth labels, demonstrating that the different modes are separated. There is a high level of correspondence between the ground truth and learned labels, although a few transitions are mislabeled.

## 6.3 FSM SYNTHESIS

We evaluate the performance of our FSM synthesis algorithm across all four environments.

For each environment, we use the expert policy to generate input data. We then use SWMPO to partition the transitions in the input data into the number of modes for that environment. For illustration purposes, we include all the labeled data for the `PointMass` environment (see fig. 5). We then synthesize the FSMWM. We compare the states visited by the synthesized FSMWM in unseen data against the ground truth states, as well as the visited states predicted by a hidden Markov model with Gaussian emissions fitted to the training data. See fig. 6). We calculate the accuracy of each partition with the Levenshtein distance to ground-truth labels (see fig. 8).

In `PointMass`, SWMPO outperforms HMMs and in `LiDAR-Racing` and `Salamander`, SWMPO significantly outperforms HMM. In `Bipedal Walker`, SWMPO marginally beats

HMM, however, where the models struggle to capture the underlying dynamics of the agent in its environment.

## 7 LIMITATIONS

The main limitations of the framework are stated formally as assumptions in section 5.1. The main assumption is that the partial observability of the environment is a consequence solely of the mode variable. Another limitation is that the mode variable must be approximated from a single transition; generalizing this to allow for modes that require multiple steps to be identified is left for future work.

## 8 CONCLUSION

We presented a novel framework for synthesizing Finite State Machine World Models (FSMWMs) in an unsupervised manner using low-level, non-visual continuous observations. We outlined the key assumptions underpinning our approach and demonstrated its applicability. Our analysis shows that the synthesized FSMWMs effectively capture the underlying structure of the environment by mapping latent modes to discrete states. Additionally, our algorithm matches or surpasses the performance of a Hidden Markov Model baseline on challenging dynamical systems. The implementation of the framework and all the code necessary to replicate the experiments, including hyperparameters, are attached to this manuscript, and are open sourced.

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

# A NOTATION TABLE

Table 1: Notation Table

| Symbol | Meaning |
|---|---|
| $\mathcal{M}$ | Markov Decision Process (MDP) |
| $S \in \mathbb{R}^k$ | Set of observations |
| $A$ | Set of actions |
| $T : S \times A \to S$ | Transition function |
| $S_0$ | Distribution of initial states |
| $R : S \times A \times S \to \mathbb{R}$ | Reward function |
| $\pi_\theta : S \to A$ | Policy parametrized by $\theta$ |
| $V_{\pi_\theta}(s)$ | Value function for state $s$ |
| $s_t$ | Source state at time $t$ |
| $s_{t+1}$ | Next state after taking action |
| $\mathcal{F}$ | Finite State Machine World Model (FSMWM) |
| $f$ | Set of environment models |
| $\delta$ | Set of mode-transition predicates |
| $s_{t+1} = f_i(s_t, u_t)$ | Predicted next observation |
| $\delta(s_t, u_t, i)$ | Mode transition function |
| $(\mathcal{M} \otimes \mathcal{F})$ | Product MDP |
| $S^\otimes$ | Augmented state space |
| $S_0^\otimes$ | Augmented initial state distribution |
| $T^\otimes$ | Transition function for product MDP |

