# OpenReview forum: "Structured World Models From Low-Level Observations"
_ICLR.cc/2025/Conference — ICLR 2025 Conference Withdrawn Submission_

### Official Review · Reviewer_7Vzg · 2024-10-26

**Soundness:** 1
**Presentation:** 2
**Contribution:** 2
**Rating:** 3
**Confidence:** 3

**Summary:**

This paper proposed an unsupervised method that can cluster agent behaviours and build finite state machines over it.

**Strengths:**

The authors have identified a meaningful research gap, and their motivation for conducting the study is compelling.
Clustering agent behaviours and building finite state machines could be useful for hierarchical RL.

**Weaknesses:**

Overall, the paper reads more like an experimental report or an essay rather than a fully developed research contribution.

1. The proposed algorithm depends heavily on the initial expert policy $\pi_0$, which limits its applicability in online learning settings.

2. While the motivation centres on improving RL control via FSM synthesis, the paper does not demonstrate how this method performs when integrated with RL algorithms. The experiments are limited to unsupervised labelling and FSM generation, which undermines the credibility of the claimed benefits for RL control.

3. Although the authors compare their approach with an HMM-based method, the details of the latter are insufficiently explained, with no descriptions or references provided. Generally speaking, this paper lacks of comparison with other methods (experimentally or analytically).

4. The caption for Figure 4 is incomplete, and there is a typographical error on line 053.

**Questions:**

See the weakness. Here are some extras:

1. How are the ground truth labels generated?

2. How much data is collected to get the results in the paper?

---

### Official Review · Reviewer_xDxB · 2024-11-01

**Soundness:** 1
**Presentation:** 1
**Contribution:** 2
**Rating:** 3
**Confidence:** 4

**Summary:**

The paper introduces Structured World Modeling from Low-Level Observations (SWMPO), a framework for unsupervised learning of neural Finite State Machines (FSMs) that represent the high-level structure of dynamic environments. SWMPO segments an environment into modes through unsupervised clustering of time-series data, synthesizing an FSM that can support tasks like policy optimization in reinforcement learning. The authors evaluate SWMPO on four environments with continuous dynamics, showcasing the method's ability to capture environmental structures.

**Strengths:**

The concept of learning environment structure through FSMs in an unsupervised manner from low-level continuous observations (such as sensor readings) is novel and could be a useful contribution to reinforcement learning and world modeling research. The paper provides detailed explanations of the SWMPO method, including algorithms and assumptions. Theoretical concepts, such as mode synthesis and transition predicate generation, are carefully formalized, supporting readers in understanding the approach. This work addresses a gap in world modeling for environments where the state dynamics can benefit from structured representations, particularly useful in scenarios with distinct modes (e.g., land vs. water). The FSM-based approach may offer advantages in certain robotic and control tasks where mode identification is crucial for efficient policy learning.

**Weaknesses:**

The paper devotes a large portion of space to formal definitions and algorithms but lacks comprehensive experimentation. Key elements, such as ablation studies on hyperparameters (e.g., epsilon, terms in the loss function, and pruning), are missing, which would help understand the impact of various components and parameters in SWMPO.

Testing SWMPO on only four environments limits its generalizability. Notably, three of these environments are custom-designed, raising questions about reproducibility and relevance. Including standardized benchmarks (e.g., MiniGrid or Four Rooms) would provide a more robust evaluation. Comparisons are limited, with only one baseline, Hidden Markov Models (HMMs). Introducing additional baselines like DreamerV3 or simpler models (e.g., behavioral cloning) would strengthen the experimental results and clarify the unique strengths of FSMs over other architectures. The results lack interpretation, and some visual elements are unclear. For instance, Figure 6d seems to illustrate a failure mode, yet this is not discussed. Similarly, Figure 4's caption is cluttered, and details like the number of seeds for Figure 8 are missing. This lack of detail reduces the clarity and interpretability of the findings.

Most importantly, the FSM's performance advantage over HMMs is sometimes minimal, particularly in Figure 8, where the FSM's results lie within HMM’s performance range in three of the four cases. This raises questions about the FSM's robustness as a general-purpose world model.

I suggest revising the paper and resubmit with a stronger experiment section, including: ablation studies on key hyperparameters and components, pruning etc., as well as broader environmental benchmarks and standardized baselines to demonstrate SWMPO's applicability across various scenarios. An expanded discussion section interpreting results, especially on observed failure modes and performance variations between SWMPO and HMM.

**Questions:**

- Could some of the definitions, algorithms, and assumptions be shifted to the appendix to allow more space for experimental analysis?
- Have you considered evaluating on additional environments such as standardized benchmarks like MiniGrid or Four Rooms?
- Why were baselines like DreamerV3 and behavioral cloning omitted from the comparisons?
- Could you provide an interpretation of failure modes, specifically the anomaly seen in Figure 6d?
- How many seeds were used in Figure 8, and what measures were taken to ensure robustness in the experiments?

---

### Official Review · Reviewer_fsLJ · 2024-11-04

**Soundness:** 2
**Presentation:** 2
**Contribution:** 1
**Rating:** 1
**Confidence:** 5

**Summary:**

The paper proposes a method for learning dynamical systems under changing dynamics, where the change is caused by additional causal variable namely mode (m). The mode m is treated as a parameterized function, of the most recent interaction history. m along with a parameterized dynamics function f (both parameterized as neural nets) are learned end to end via a traditional objective function for dynamics learning with a mutual information-based regularization. Experiments which mainly focus on detecting the modes are benchmarked on a variety of non-stationary benchmarks (possibly for one-step ahead predictions). The method performs well compared to an HMM baseline.

**Strengths:**

These are the strengths of the paper in my opinion.

1. Targets an important problem of learning dynamics under non-stationary dynamics.
2. Interesting method, and the additional regularization with mutual information seems novel especially in the context of dynamics learning under non-stationarity.
3. The algorithm seems simple as far as computational complexity and implementation are concerned.

**Weaknesses:**

These are the weaknesses of the paper in my opinion.

1. The paper ignores a plethora of literature in this domain, like switching state-space models[1][2], hidden-paramter SSMs[3] etc which tackles the same problem but with more principled approaches and in more complex settings. These should be made as baselines.

2. The main purpose of using this as a world model, is for making action conditional multi-step ahead predictions. No experiments on this regard are seen.

3. The paper could be made even more stronger, if control experiments (model-based RL) with this world model are performed.

4. There are a few typos, which need to be corrected, including in the objective function in equation (1). It should be $
f\left(m\left(o_{t-1}, a_{t-1}, o_t\right), a_t, o_t \right)$ in Equaton 1? what is e and d ?

**Questions:**

1. Would this method scale to partially observable domain like images ?
2. Can you provide multi step ahead prediction comparison ?
3. Maybe add more baselines like [1],[2] ?
4. An ablation on the importance of mutual information regularization scheme would be very interesting.


References
1. Switching SSM (one among many) https://proceedings.neurips.cc/paper/2020/file/aa1f5f73327ba40d47ebce155e785aaf-Paper.pdf
2. Hidden Parameter Recurrent State Space Models For Changing Dynamics Scenarios  https://arxiv.org/abs/2206.14697

---

### Official Review · Reviewer_CDcn · 2024-11-08

**Soundness:** 2
**Presentation:** 1
**Contribution:** 3
**Rating:** 5
**Confidence:** 4

**Summary:**

This paper introduces SWMPO (Structured World Modeling From Low-Level Observations), a framework for unsupervised learning of neural Finite State Machines (FSMs) to represent environment structure. The proposed method operates on continuous, low-level, non-visual data (e.g., LiDAR, joint positions) and involves three key steps: 1) labeling transitions in a dataset based on learned mode embeddings, 2) pruning spurious transitions between modes, and 3) synthesizing transition predicates between modes. The authors evaluate SWMPO on four simulated environments and demonstrate improved performance over Hidden Markov Models (HMMs) in most cases.

**Strengths:**

- The application of FSMs for environment modelling is an interesting problem
- The method described appears to be general and well motivated
- SWMPO performs well in the experiments presented.

**Weaknesses:**

- The paper makes references (L158, L166, L180, L344) to how SWMPO has been used in the paper in an RL training loop, but there are no experiments to back up that claim. Either these experiments should be added or the claims removed from the paper.
- All the experiments assume access to an expert policy.
- Assumption 2 is restrictive. It is not trivial to assume that a POMDP can become a deterministic MDP conditioned on the mode.
- There are only two environments tested where the number of modes is >2 and LiDAR racing appears to have simple dynamics. Most environments where this system can be used have more than two modes.
- The paper assumes knowledge of the number of modes in the environment.
- Presentation is poor, making the paper difficult to read (see changes requested in the section below)

I believe the paper shows promise but in its current state is not ready for acceptance. I would be willing to reconsider if the authors are able to provide the requested changes.

**Questions:**

- How would SWMPO perform if the data was from a suboptimal/random policy? Is an expert policy a requirement for the algorithm?
- Could you replicate Fig. 7 with the BipedalWH or LiDAR environments. That would be more convincing since they have more modes to partition.
- What are the observations given to SWMPO in each of the environments?
- How much data was given for each environment?
- What is the dimension of the mode vector?
- What are the modes in the LiDAR racing and BipedalWH environments?
- What is the value of $\epsilon$ used for pruning? Is the algorithm sensitive to this hyperparameter?

**Presentation and grammatical changes**
- The optimization problem in Eq. 1 is defined over $e$ and $d$ which are never defined. I am assuming that these should be $m$ and $f$. Similar error in Algorithm 2.
- Eq. 1 is missing an $o_t$ as input to $f$. Also does it assume access to the environment state transition function $T$? Shouldn't it be the difference in observations and not states as per L205.
- L143, $o$ should belong to the space of observations $\Omega$ not states.
- Algorithm 1 does not seem to be necessary since it just consists of calling one function.
- Please also label the colors used to describe the modes in Figs 5 and 6.
- In Fig 4, the caption is cut off.
- The text in the figures (for eg 5 and 6) is extremely small, and is illegible

---

### Note · Authors · 2024-11-19

I have read and agree with the venue's withdrawal policy on behalf of myself and my co-authors.